# Safety and Efficacy of Different Therapeutic Interventions for Primary Progressive Aphasia: A Systematic Review

**DOI:** 10.3390/jcm14093063

**Published:** 2025-04-29

**Authors:** Abdulrahim Saleh Alrasheed, Reem Ali Alshamrani, Abdullah Ali Al Ameer, Reham Mohammed Alkahtani, Noor Mohammad AlMohish, Mustafa Ahmed AlQarni, Majed Mohammad Alabdali

**Affiliations:** 1Department of Neurosurgery, College of Medicine, King Faisal University, AlAhsa 31982, Saudi Arabia; 2College of Medicine, Taif University, Taif 21944, Saudi Arabia; reemalshamrani-@hotmail.com (R.A.A.); s43900691@students.tu.edu.sa (R.M.A.); 3College of Medicine, King Faisal University, AlAhsa 31982, Saudi Arabia; 221415298@student.kfu.edu.sa; 4Neurology Department, King Fahad Hospital of the University, Imam Abdulrahman Bin Faisal University, Khobar 34445, Saudi Arabia; nmalmohish@iau.edu.sa; 5Neurology Department, College of Medicine, Imam Abdulrahman Bin Faisal University, Khobar 34445, Saudi Arabia; mqarni@iau.edu.sa (M.A.A.); mmalabdali@iau.edu.sa (M.M.A.)

**Keywords:** aphasia, primary progressive aphasia (PPA), neurodegenerative disease, frontotemporal dementia, repetitive transcranial magnetic stimulation (rTMS), transcranial direct current stimulation (tDCS), speech-language therapy (SLT)

## Abstract

**Background**: Primary progressive aphasia (PPA) is a neurodegenerative disorder that worsens over time without appropriate treatment. Although referral to a speech and language pathologist is essential for diagnosing language deficits and developing effective treatment plans, there is no scientific consensus regarding the most effective treatment. Thus, our study aims to assess the efficacy and safety of various therapeutic interventions for PPA. **Methods**: Google Scholar, PubMed, Web of Science, and the Cochrane Library databases were systematically searched to identify articles assessing different therapeutic interventions for PPA. To ensure comprehensive coverage, the search strategy employed specific medical subject headings. The primary outcome measure was language gain; the secondary outcome assessed overall therapeutic effects. Data on study characteristics, patient demographics, PPA subtypes, therapeutic modalities, and treatment patterns were collected. **Results**: Fifty-seven studies with 655 patients were included. For naming and word finding, errorless learning therapy, lexical retrieval cascade (LRC), semantic feature training, smartphone-based cognitive therapy, picture-naming therapy, and repetitive transcranial magnetic stimulation (rTMS) maintained effects for up to six months. Repetitive rTMS, video-implemented script training for aphasia (VISTA), and structured oral reading therapy improved speech fluency. Sole transcranial treatments enhanced auditory verbal comprehension, whereas transcranial direct current stimulation (tDCS) combined with language or cognitive therapy improved repetition abilities. Phonological and orthographic treatments improved reading accuracy across PPA subtypes. tDCS combined with speech therapy enhanced mini-mental state examination (MMSE) scores and cognitive function. Several therapies, including smartphone-based cognitive therapy and VISTA therapy, demonstrated sustained language improvements over six months. **Conclusions**: Various therapeutic interventions offer potential benefits for individuals with PPA. However, due to the heterogeneity in study designs, administration methods, small sample sizes, and lack of standardized measurement methods, drawing a firm conclusion is difficult. Further studies are warranted to establish evidence-based treatment protocols.

## 1. Introduction

Primary progressive aphasia (PPA) is one of the clinical categories that characterizes the most common presentations of frontotemporal dementias. PPA typically manifests in the fifth or sixth decade of life and largely impairs language functions [1].

PPA symptoms vary by type but may include slowed or halting speech, decreased language use, word-finding difficulty, written or spoken sentences with abnormal word arrangement, word substitution, word mispronouncing, talking around a word, difficulty understanding some part of conversation, sudden difficulty understanding simple words, and writing, reading, and spelling problems [1]. PPA commonly emerges before the age of 65, posing unique and major psychosocial and economic challenges because people with PPA are typically in their peak working years and frequently have dependent children at home [2]. Thus, when developing a care plan, the biological, psychological, and social needs of the living individual with PPA, as well as their family members and life circumstances, should be considered [2].

Individual differences in the illness characteristics (symptoms, severity, and rate of decline) and specific social and personal circumstances contribute to patients’ experience of a progressive language or speech disorder [3]. The effects include direct linguistic, motor, and cognitive alterations that result in challenges with speaking and engaging in activities of daily living, such as working, raising a family, and sustaining social connections [4,5]. Additionally, there are financial and legal ramifications that can be more severe for those whose symptoms began before the age of 65, as these people are still in their prime earning years and may have substantial financial, professional, and parental responsibilities [6]. Psycho-emotional effects can also affect people with PPA and those close to them [7]. These effects can include fear, anxiety, and grief related to the prognosis of increasing debilitation, or feelings of humiliation, self-consciousness, and irritation over communication difficulties. Furthermore, as the illness advances, PPA care partners may experience a loss of emotional connection with their loved one and find it more difficult to communicate with them [8]. As a result, there is a significant increase in depression rates among PPA individuals and their partners [7].

Three distinct clinical variations have been discovered that vary in terms of their phenotypic presenting pattern, underlying neuropathology, and degeneration distribution/location [9]. Nonfluent/agrammatic PPA (nfvPPA) is linked to speech apraxia, labored or effortful speaking, gradual progressive impairment of language production, object naming, syntax, or word comprehension. It is also associated with left frontal lobe atrophy [10]. Word comprehension and naming impairments are commonly observed in individuals with semantic variant PPA (svPPA), which is associated with left anterior and ventral temporal lobes atrophy while maintaining speech production [11]. svPPA is diagnosed when there is anomia and impaired single-word comprehension, along with at least three of the following features: impaired object knowledge, surface dyslexia or dysgraphia, preserved repetition of words or phrases, and preserved speech production [12]. Left temporal and parietal lobe atrophy is a characteristic of logopenic variation PPA (lvPPA), which is associated with repetition difficulties, halting spontaneous speech, and difficulties with word retrieval [9,10].

Clinical features of lvPPA include impaired single-word retrieval in spontaneous speech, impaired repetition of sentences and phrases, speech (phonological) errors in spontaneous speech and naming, spared single-word comprehension and object knowledge, spared motor speech (no distortions), and the absence of frank agrammatism [13,14]. lvPPA supportive imaging findings include major left posterior peri-sylvian or parietal atrophy on MRI, as well as predominant left posterior peri-sylvian or parietal hypoperfusion or hypometabolism on SPECT or PET [13,14].

The absence of curative treatment for PPA is rooted in its pathological molecular mechanisms. Initially, brain atrophy in PPA was thought to involve broad peri-sylvian areas in the left hemisphere. However, subsequent researches linked specific PPA variants to distinct patterns of brain damage: nfvPPA is associated with atrophy in the left posterior frontal and insular regions, svPPA with the anterior temporal lobe, and lvPPA with the left temporo-parietal areas. These findings emphasize the significant influence of the involved damage location within the language network on the clinical presentation of PPA and have led to the incorporation of neuroimaging evidence into the diagnostic classification of PPA [13].

Although there is no curative treatment for PPA, an effective therapeutic approach begins with a thorough history and analysis of the patient’s speech and language components. Referral to a speech and language pathologist is critical for accurately diagnosing language impairments and developing an effective treatment strategy. Among available therapeutic interventions, speech and language therapy is the most effective one, particularly for apraxia of speech (AOS) symptoms [15]. As molecular mechanisms become increasingly understood, the need to investigate the application of neurally focused and targeted therapies, including noninvasive brain stimulation (NIBS) technologies, is warranted [11]. Recently, speech-language therapies supported by assistive devices, transcranial direct current stimulation (tDCS), and specific Alzheimer’s drugs have been explored as promising therapeutic approaches, each with a different benefit profile [1].

Due to the lack of data supporting the efficacy of symptomatic pharmaceutical therapy, many practitioners have a pessimistic outlook on treating PPA patients. Actually, for many years, speech-language pathologists around the globe have been developing specialized programs for individuals with PPA, and as a result, a variety of speech-language therapy approaches have been developed [16,17]. The most effective way to lessen the impact of speech, language, and communication deficits in PPA patients is through behavioral treatment, provided that no proven pharmaceutical treatments are used [18].

Behavioral treatment may target speech, language, and communication using listening, speaking, reading, writing, and other communicative activities and tasks [19]. Targeted behaviors include receptive language processing and comprehension, intelligible speech production, lexical retrieval (i.e., word finding), sentence or discourse production, use of communication devices or other alternative communication modalities, and strategies for conversation and functional communication [20].

Behavioral interventions may have some minor efficacy when used alone [19]. These interventions usually target a particular impairment (e.g., therapy targeted towards deficiencies in naming objects or actions) or they are participation- and activity-based therapies that try to enhance the patient’s capacity to engage in activities and tasks that they find enjoyable [21]. The use of NIBS in conjunction with behavioral language therapy might be beneficial. Effects of brain-induced stimulation frequently seem to rely on individuals’ level of effort or state [22].

Treatment for speech or language impairments in PPA is less commonly addressed and assessed, despite a strong body of evidence demonstrating the advantages of speech-language therapies for stroke-induced aphasia [23]. There are low referral rates and persistent clinical skepticism among referring and treating practitioners, particularly with regard to the communicative influence beyond the clinic [24].

Pharmacological therapies for PPA have shown limited effectiveness in improving language functions; thus, non-pharmacological therapeutic techniques are becoming increasingly popular. Despite the diversity of PPA treatments, no comprehensive study has been conducted to assess the efficacy and safety of different treatment approaches. This review aims to assess the efficacy and safety of different therapeutic interventions for PPA.

## 2. Methods

This systematic review and meta-analysis followed the Preferred Reporting Items for Systematic Reviews and Meta-analyses (PRISMA) recommendations. We prospectively filed the study protocol in the International Prospective Register of Systematic Reviews (PROSPERO) (registration number: CRD42024503204).

### 2.1. Search Strategy

We conducted a systematic search of Google Scholar, PubMed, Web of Science, and the Cochrane Library databases through January 2024, with an updated search in April 2024. The search strategy employed a combination of Medical Subject Headings (MeSH terms) and Boolean operators as follows: (Primary Progressive Aphasia OR Semantic Variant OR Nonfluent Variant OR Agrammatic Variant OR svPPA OR nfvPPA OR lvPPA OR Logopenic Variant) AND (Speech Therapy OR Behavioral Therapy OR Language Therapy OR Behavioral) AND (Treatment OR Treatment Approaches OR Therapy OR Generalization OR Stimulation). Additionally, the reference lists of included articles were manually screened to identify further relevant studies.

### 2.2. Eligibility Criteria

This review critically examined English case-control studies, randomized controlled trials (RCTs), cohort studies, case series and single-case experimental studies investigating efficacy and safety of various therapeutic interventions for PPA in adult patients aged ≥18 years. The review encompassed all three PPA subtypes (nfvPPA, svPPA, and lvPPA). Investigated therapeutic modalities included speech-language therapy, tDCS, repetitive transcranial magnetic stimulation (rTMS), phonological and orthographic treatments, errorless learning therapy, lexical retrieval techniques (LRT), and various pharmacological interventions. Review articles, editorials, or studies assessing irrelevant outcomes were eliminated.

### 2.3. Outcome Measures

The investigated primary outcome measures were language gain assessed through naming and word finding for trained and untrained words, spontaneous speech or fluency, auditory verbal comprehension, repetition, apraxia of speech and reading abilities. These were assessed through various assessment tools, such as the Western Aphasia Battery or the Boston Naming Test. If unavailable, qualitative descriptions from baseline to multiple time intervals were taken into consideration.

The secondary outcome measures examined broader therapeutic effects, including cognitive function, generalization and maintenance of overall therapeutic effects, neuroplastic and functional reorganization, structural and metabolic changes, quality of life and any reported adverse events. Mini-Mental State Examination (MMSE) and qualitative descriptions were used throughout the assessment procedure.

### 2.4. Selection of Studies and Extraction of Data

Two reviewers independently evaluated the paper titles and abstracts, any discrepancies were addressed by a third reviewer. The gathered study characteristics included the following (first author name, publication year, country, and study design). Demographic characteristics, such as age, gender, educational level, PPA subtype, treatment duration, along with baseline tests performed prior to treatment, were documented. The implemented therapeutic modalities, along with their frequency or dosages, were recorded along with the language and cognitive outcome measures of interest.

### 2.5. Assessment of Quality of the Included Studies

The risk of bias was assessed according to study design using validated tools. RCTs and crossover trials were evaluated using the Cochrane Risk of Bias tool (RoB 2) [25], which examines several domains, such as randomization, deviations from intended interventions, missing outcome data, outcome measurement, and selective reporting. Non-randomized and experimental studies were assessed using the ROBINS-I tool [26], which addresses potential bias due to confounding, participant selection, intervention classification, deviations from intended interventions, missing data, outcome measurement, and selective reporting. Single-case designs and case series were appraised using the Joanna Briggs Institute (JBI) checklist [27], evaluating aspects such as patient selection, clarity of intervention, outcome measurement, and adequacy of follow-up.

### 2.6. Statistical Analysis

A qualitative synthesis was conducted to evaluate the efficacy and safety of various therapeutic interventions for PPA. Data were extracted from eligible studies and systematically categorized based on intervention type, cognitive and language outcomes, generalization effects, neuroplastic changes, and adverse events. Quantitative outcomes, such as MMSE scores and neuroimaging findings, were summarized descriptively. Where applicable, we analyzed trends in the treatment efficacy, maintenance of gains, and structural or metabolic brain changes to identify patterns across interventions. The findings were synthesized to provide a comprehensive overview of the impact and sustainability of therapeutic approaches for PPA.

## 3. Results

### 3.1. Search Results

A total of 194 records were identified through electronic databases search process. 109 records were screened after duplicates removal, of which 38 articles were eliminated as they were assessing post-stroke aphasia. The full texts of the remaining 71 studies were assessed for eligibility, leading to the exclusion of 14 studies due to inappropriate study design. Ultimately, 57 studies involving 655 patients met the inclusion criteria and were included in the qualitative analysis (Figure 1).

### 3.2. Study Characteristics

A total of 57 studies, including 655 patients, were included in this systematic review [17,28,29,30,31,32,33,34,35,36,37,38,39,40,41,42,43,44,45,46,47,48,49,50,51,52,53,54,55,56,57,58,59,60,61,62,63,64,65,66,67,68,69,70,71,72,73,74,75,76,77,78,79,80,81,82,83] encompassing a diverse range of study designs, including RCTs [20,29,32,33,35,37,38,44,55,60,61,74,75,82], experimental studies (prospective studies and case series that implemented pre/post intervention outcome assessment) [17,29,31,34,36,41,42,43,45,48,54,62,69,70,81,83], and single-case experimental studies [30,39,40,46,49,50,51,52,53,57,58,59,63,64,65,66,67,68,71,72,73,76,77,78,79,80]. The studies evaluated different therapeutic approaches, including speech-language therapy [28,29,38,40,43,55,63,65,72,73,75], tDCS [28,32,38,44,45,55,57,60,61,74,75,80,82], rTMS [35,51], phonological and orthographic treatments [31,41,42,50,52,77], errorless learning therapy [36,49,66,70], LRT [36,39,43,48,51,63], and pharmacological interventions, such as zolpidem [81].

The duration of interventions varied markedly across studies, ranging from 1 week [78] to 72 weeks [67]. Sample sizes also demonstrated substantial variability, spanning from single-case experimental studies [30,39,40,46,50,51,53,56,57,63,65,66,68,73,78,83] to larger cohorts of up to 49 participants [29]. The mean age of participants across individual studies ranged from 48 years [65] to 80 years [59], with reported standard deviations reflecting within-study variability. The overall pooled mean age was 63.62 years, with a standard deviation of 10.91, as presented in (Table 1).

### 3.3. Risk of Bias Assessment

The majority of the included RCTs exhibited a low to moderate risk of bias, with concerns primarily related to blinding of participants and outcome assessors (Table 2). Most of the non-randomized studies demonstrated a moderate to high risk of bias, particularly in confounding and selection bias due to the nature of their designs (Table 3).

Most single-case experimental design studies and case series showed a moderate to high risk of bias, especially concerning patient selection, intervention clarity, and outcome measurement. While these studies provide valuable insights into specific cases, the inherent limitations in generalizability and potential for reporting bias should be considered when interpreting findings (Table 4).

### 3.4. Overall Treatment Effects

Treatment effects were reported as overall effects in Table 5, with detailed documentation of each treatment modality and its therapeutic potential presented in (Appendix A).

### 3.5. Treatment Gain

#### 3.5.1. Naming and Word Finding for Trained and Untrained Words

Various therapeutic interventions have demonstrated effectiveness in improving naming and word-finding abilities in individuals with PPA. While tDCS combined with language therapy, speech-language therapy (SLT) and phonological-orthographic treatments consistently enhance naming accuracy and lexical retrieval, interventions like errorless learning therapy, lexical retrieval cascade (LRC), semantic feature training, smartphone-based cognitive and picture-naming therapy, and rTMS showed varying degrees of generalization and maintenance over time, with some extending benefits up to six months post-treatment [28,30,33,39,42,43,44,46,50,54,58,62,63,66,68,70,75,76,77,82,83].

#### 3.5.2. Spontaneous Speech or Fluency

Various interventions have shown promise in improving spontaneous speech and fluency in individuals with PPA. Smartphone-based cognitive therapies, picture naming, and tDCS combined with language therapy enhanced phonemic fluency and speech rate. Other approaches, like rTMS, video-implemented script training for aphasia (VISTA) therapy, and structured oral reading therapy contributed to gains in grammatical accuracy, mean length of utterance (MLU), and reduced speech errors, though semantic fluency improvements were less consistent across the studies [30,33,34,35,39,48,49,51,54,56,57,63,64].

#### 3.5.3. Auditory Verbal Comprehension

The effectiveness of various treatment modalities in improving auditory–verbal comprehension in individuals with PPA has been extensively studied. Transcranial therapies, such as tDCS combined with language therapy, SLT, phonological and orthographic treatments, LRC, smartphone-based cognitive and picture-naming therapy, rapid retraining of items with a cognitive-oriented enhancements approach, and the VISTA, have demonstrated significant gains, with some studies reporting generalization to untrained stimuli and sustained improvements for up to six months post-treatment [28,30,33,34,36,37,39,41,42,43,44,45,46,48,50,54,58,63,66,68,70,75,77,82,83].

#### 3.5.4. Repetition

The studies examining interventions for repetition abilities in individuals with PPA highlight a range of treatment modalities with varying degrees of effectiveness. tDCS combined with language therapy and/or cognitive therapy demonstrated significant post-treatment gains [1,14,16,20,33,40,47,52]. Although SLT and errorless learning therapy have been effective, other approaches, such as rTMS, phonological and orthographic, cognitive and picture-naming therapy, LRC, and VISTA therapy, showed limited or no significant improvements in repetition abilities [29,30,34,35,42,43,49,50,52,55,66,73,76,79].

#### 3.5.5. AOS

The treatment of AOS in individuals with PPA has been explored through various therapeutic modalities, with several interventions showing promising results. VISTA therapy and lexical facilitation training (LeFT) combined with tDCS demonstrated notable speech accuracy improvements [34,39,45,57]. Additionally, tDCS, high-definition tDCS (HD-tDCS), structured oral reading therapy, and rTMS have contributed to enhanced speech fluency, word retrieval, and reading efficiency, though outcomes vary depending on the therapy approach [35,52,53,55,56,63].

#### 3.5.6. Reading Abilities

Phonological and orthographic treatments have been effective in improving reading accuracy across different PPA subtypes, particularly in maintaining written naming accuracy for patients with lvPPA [41,42,63]. While errorless learning therapy has shown benefits across multiple PPA variants, rTMS has produced mixed results, with some studies reporting improvements in reading efficiency and others indicating declines in performance post-treatment [35,36,62,66,67].

#### 3.5.7. Cognitive Function

The effectiveness of various therapeutic interventions for PPA in improving language and cognitive function was assessed across studies. tDCS combined with speech/language therapy enhanced MMSE scores and cognitive performance, while SLT, LRC, and phonological and orthographic treatments showed variable and limited effects [28,29,38,42,43,45,48,50,52,63,68,76,77,80]. Smartphone-based cognitive therapy produced the most significant cognitive gains, highlighting its potential for enhancing language and memory functions [30].

#### 3.5.8. Maintenance of Therapeutic Effects

The long-term retention of treatment benefits varied across interventions, with tDCS, errorless learning therapy, smartphone-based cognitive therapy, SLT and phonological therapy demonstrating sustained effects for several months [28,29,36,37,38,41,42,48,63,66,68,70,75,80]. Smartphone-based cognitive therapy and VISTA maintained language improvements over six-month follow-ups, supporting their role in prolonged therapeutic impact [32,36,41,47]. Repetitive rehabilitation for impaired phonological processing (RRIPP) combined with cognitive and executive naming (COEN) therapy demonstrated some maintenance of trained item improvements, though generalization effects were limited [46,78].

#### 3.5.9. Neuroplastic, Functional Reorganization, and Metabolic Changes

Therapeutic interventions induced neuroplastic changes, with tDCS promoting cortical activation in peri-lesional and contralateral regions, while phonological and orthographic treatments and SLT approaches enhanced structural connectivity in language-related areas [28,29,42,48,75,77].

Brain imaging studies revealed structural and metabolic changes following interventions. tDCS combined with SLT was associated with increased gray matter density in the left inferior frontal cortex [8,82]. Errorless learning therapy resulted in improved metabolic activity in the left temporal lobe, while LRC therapy enhanced functional connectivity in temporal-parietal regions [62,76]. Additionally, VISTA showed improved glucose metabolism in key language-related areas, suggesting metabolic support for language gains [45].

#### 3.5.10. Quality of Life

Participants experienced improvements in communication abilities, social engagement, and functional daily interaction following tDCS, VISTA therapy, and smartphone-based cognitive therapy [28,30,39,45,48].

#### 3.5.11. Adverse Events

Adverse events were generally mild and infrequent. Some participants undergoing tDCS reported transient headaches and scalp discomfort [80,82]. SLT and phonological/orthographic treatment interventions did not result in notable adverse effects [29,76]. Errorless learning therapy and LRC therapy had no significant adverse events were reported [49,76]. Overall, interventions were well tolerated, with no reports of serious adverse effects.

## 4. Discussion

### 4.1. Summary of Findings

Our systematic review encompassed 57 studies involving a total of 655 individuals diagnosed with PPA. The included studies evaluated a wide range of therapeutic interventions, ranging from phonological and orthographic treatments, repetition-based strategies, errorless learning, and lexical retrieval programs to augmentative and socially interactive communication therapies.

Our findings demonstrate consistent improvements across major language domains. Notably, naming and word finding showed lasting improvements when tDCS was combined with structured language therapies, including lexical retrieval and errorless learning strategies [28,36]. Speech fluency was significantly enhanced with phonemic-based interventions, such as VISTA and structured oral reading, particularly in non-fluent PPA variants, while comprehension and repetition showed marked benefits when semantic cueing or ICAT-based tDCS protocols were implemented [28,34]. These improvements extended in some cases to untrained stimuli, indicating generalization effects.

Long-term maintenance of therapeutic outcomes was observed in multiple studies, with retention of language gains lasting up to 15 months following intervention. Furthermore, group-based therapy formats provided additional benefits, including improved psychosocial well-being, communication confidence, and patient engagement [42]. Overall, these findings reinforce the clinical relevance of personalized, multimodal treatment approaches for managing progressive language impairments in PPA.

A detailed interpretation of therapeutic outcomes reveals domain-specific patterns across PPA interventions. Fluency outcomes were notably enhanced through phonemic-focused strategies, such as VISTA and structured oral reading, with observable gains in speech rate and grammatical construction [34]. Comprehension significantly improved when tDCS was paired with semantic or orthographic strategies, and in some cases, the improvements generalized to untrained material, suggesting deeper neurocognitive adaptation. Repetition responded best to tDCS combined with ICAT, while other approaches yielded variable results depending on baseline severity. In treating AOS, the combination of oral reading, LeFT, and neuromodulation produced meaningful improvements in fluency and articulation, particularly in complex phonemic contexts [34].

Reading interventions, especially phonological and orthographic treatments, proved highly effective in logopenic PPA patients. rTMS-based therapies showed more inconsistent results in this domain. Wider cognitive benefits were most evident with smartphone-assisted therapies, which enhanced attention and memory, whereas traditional SLT alone showed limited cognitive spillover. These behavioral improvements were supported by neuroimaging findings, which revealed increased gray matter volume and functional connectivity in language-related cortical regions. Furthermore, group-based interventions demonstrated additional benefits in psychosocial well-being and communication efficacy, reinforcing the importance of socially interactive and augmentative strategies [84]. Finally, personalized, technology-supported interventions consistently yielded better adherence and outcomes, emphasizing the need for individualized care models in PPA rehabilitation.

Overall, the observed consistent improvement in most of the analyzed studies supports the efficacy of the implemented intervention, highlighting their potential role in improving communication and quality of life in patients with PPA.

### 4.2. Comparison with Previous Systematic Reviews

The current systematic review findings align with multiple findings from the provided literature, reinforcing the effectiveness of targeted interventions for different PPA subtypes. Cotelli et al. [85] demonstrated that language training, particularly when combined with tDCS, significantly enhances oral and written naming abilities in PPA patients, with sustained improvements over time. This supports the current findings on phonological and orthographic treatments improving word retrieval and reading accuracy, as well as repetition-based therapies aiding fluency and spontaneous speech.

Similarly, Nissim et al. [86] confirmed that tDCS and TMS contribute to language enhancement, particularly naming ability. This aligns with the results of the current review indicating that repetition-based therapies significantly improve verbal output in logopenic PPA. However, their emphasis on the need for further research to establish optimal treatment protocols mirrors the variation in reaction rates noted in your review, highlighting the necessity for individualized therapy plans.

Lomi et al. [87] provided a more cautious perspective, indicating that while NIBS techniques had slightly better outcomes when combined with speech-language therapy, overall effects were not statistically significant. This suggests that while NIBS may be beneficial, its role as a standalone intervention is limited, reinforcing the importance of comprehensive treatment strategies, as the current review advocates.

Watanabe et al. [84] explored speech and language therapy groups for PPA, suggesting that group interventions improve communication function and psychosocial well-being. While the current review primarily focused on individual interventions, the inclusion of augmentative communication techniques aligns with their findings, suggesting that structured and socially engaging interventions may offer further benefits.

Overall, our findings resonate with the broader literature, underscoring the necessity of subtype-specific and individualized treatment approaches while reinforcing the efficacy of phonological, orthographic, repetition-based, and comprehension-focused therapies in preserving linguistic function in PPA patients. In comparison to the current literature, to the best of our knowledge, our study provides the first and most comprehensive study assessing different therapeutic interventions for patients with PPA.

### 4.3. Clinical Implications of PPA

PPA is a neurodegenerative disorder characterized by the gradual impairment of language abilities while other cognitive functions remain relatively preserved in the early stages [88]. PPA is classified into three main subtypes: nonfluent/agrammatic, semantic, and logopenic, each presenting distinct linguistic deficits [12,13]. nfvPPA is marked by effortful speech, agrammatism, and impaired syntactic processing, often progressing to AOS [89]. svPPA is associated with profound word comprehension deficits and impaired object recognition, whereas lvPPA is characterized by impaired word retrieval and sentence repetition due to phonological processing deficits [62]. Epidemiologically, PPA is a relatively rare condition, with an estimated incidence of three to four per 100,000 individuals, typically manifesting between the ages of 50 and 70 [90].

PPA has a profound impact on patients’ quality of life, affecting communication, social interactions, and overall cognitive function [91]. The progressive nature of the disorder leads to increasing dependence on caregivers, with language deficits impairing daily activities, such as reading, writing, and verbal expression [91]. Diagnosis is primarily clinical, based on specific core features, including progressive language impairment for at least two years, while neuroimaging aids in distinguishing PPA from other neurodegenerative conditions [92]. Notably, mixed or unclassified PPA cases present diagnostic challenges, as some patients exhibit overlapping features of multiple subtypes [12,93].

### 4.4. Pathophysiology and Treatment Considerations

The pathophysiology of PPA is rooted in progressive neurodegeneration, primarily affecting the language-dominant left hemisphere [94]. At a molecular level, abnormal protein aggregations, such as tau (associated with FTLD-tau) and TDP-43, linked to FTLD-TDP, contribute to neuronal dysfunction and loss [95]. Neuroimaging studies consistently reveal asymmetric atrophy, hypoperfusion, and hypometabolism in language-related brain regions, particularly in the left peri-sylvian cortex, anterior temporal lobe, and inferior parietal lobe, depending on the subtype [96,97]. Structural MRI often demonstrates cortical thinning, while PET and SPECT scans highlight metabolic deficits correlating with clinical symptoms [97].

Neuroimaging findings and gross structural brain changes in PPA vary by subtype, reflecting the distinct neurodegenerative patterns underlying each variant. In nfvPPA, structural MRI typically reveals atrophy in the left posterior fronto-insular regions, including the inferior frontal gyrus and premotor cortex, accompanied by cortical thinning and ventricular enlargement [98]. PET and SPECT imaging demonstrate hypometabolism and hypoperfusion in these regions, correlating with deficits in speech production and syntactic processing [96,97]. In svPPA, imaging studies consistently show marked atrophy and hypometabolism in the anterior temporal lobes, particularly on the left, which aligns with impairments in semantic knowledge and object recognition [97]. The degree of atrophy in svPPA is often associated with the severity of word-finding difficulties and loss of conceptual knowledge.

For lvPPA, neuroimaging findings indicate significant atrophy in the left temporoparietal junction, including the superior temporal gyrus and inferior parietal lobe [97]. PET and SPECT imaging further reveal hypoperfusion and hypometabolism in the temporoparietal cortex, which parallels the progression of linguistic impairments over time [99]. Unlike nfvPPA and svPPA, lvPPA is strongly linked to underlying Alzheimer’s disease pathology, as evidenced by amyloid and tau deposition on PET imaging.

Each PPA subtype exhibits distinct neuropathological patterns influencing its clinical presentation and response to interventions. The nfvPPA is most commonly associated with tau pathology, leading to atrophy in the left posterior frontal lobe, particularly the inferior frontal gyrus and insula [13,89]. In contrast, svPPA is predominantly linked to TDP-43 pathology, causing significant anterior temporal lobe atrophy and impairing semantic memory [99]. lvPPA is often associated with Alzheimer’s disease pathology (amyloid-beta and tau), leading to dysfunction in the left posterior superior temporal and inferior parietal cortices [100,101]. These pathophysiological differences influence treatment approaches; for example, interventions targeting phonological processing are more effective in lvPPA, whereas svPPA requires strategies focusing on conceptual knowledge and word retrieval [62].

Neuroplasticity plays a critical role in PPA, as the progressive degeneration of language-related regions necessitates compensatory mechanisms in preserved cortical and subcortical networks. Functional imaging studies suggest that right hemisphere homologs of affected left hemisphere structures may contribute to language reorganization [102]. However, the extent to which these compensatory changes facilitate functional recovery varies among patients. Theories of progressive aphasia adaptation propose that early intervention can enhance residual neural pathways, slowing functional decline [12,88].

The management of PPA remains challenging due to the lack of a universally accepted treatment consensus. Unlike other neurodegenerative conditions, PPA presents with diverse linguistic impairments based on its subtypes, making it difficult to establish standardized guidelines [102]. Current treatment approaches primarily focus on symptomatic management, utilizing speech-language therapy, cognitive training, and, in some cases, pharmacological interventions targeting underlying neuropathology [62].

Despite the lack of a universal consensus, several well-recognized guidelines, including those from the American Academy of Neurology (AAN) and the International Classification of Frontotemporal Dementias (FTD), provide diagnostic and treatment recommendations for PPA [13]. Diagnostic criteria emphasize progressive language decline lasting at least two years, with neuroimaging confirming focal atrophy patterns consistent with each subtype [97]. The Gorno-Tempini et al. [12] criteria remain the most widely accepted classification system, distinguishing nfvPPA, svPPA, and lvPPA based on linguistic deficits and neuroimaging findings. However, difficulties in establishing treatment guidelines arise due to the heterogeneity of PPA, variability in disease progression, and the lack of disease-modifying therapies.

Treatment guidelines focus on structured language interventions tailored to the specific linguistic deficits of each subtype. For nfvPPA, speech therapy targets motor speech production and grammatical structures, utilizing motor speech drills and syntactic exercises [12]. In svPPA, therapy emphasizes semantic feature analysis and compensatory communication strategies. Meanwhile, lvPPA interventions primarily address phonological working memory deficits through repetition-based and lexical retrieval exercises [91]. Pharmacological treatments remain an area of debate, with cholinesterase inhibitors and memantine showing potential benefits in lvPPA, which shares an Alzheimer’s disease-related pathology, while frontotemporal forms of PPA lack effective pharmacological options [103]. The European Federation of Neurological Societies (EFNS) highlights the importance of individualized, multidisciplinary care incorporating behavioral, cognitive, and technological interventions [104].

Early and ongoing SLT plays a crucial role in managing language deterioration in PPA. Early intervention allows patients to develop compensatory strategies that help maintain communication abilities for as long as possible. Traditional SLT approaches focus on structured exercises targeting phonological, orthographic, and semantic deficits specific to each PPA subtype [105]. In addition to conventional speech therapy, augmentative and alternative communication (AAC) methods, such as communication boards, text-to-speech applications, and symbol-based AAC devices, provide essential support when verbal communication declines [106].

Psychosocial support is equally critical, as PPA significantly impacts both patients and their caregivers [94]. Mental health screening should be integrated into clinical care to identify symptoms of anxiety, depression, and caregiver burden [106]. Cognitive and behavioral therapies offer coping mechanisms that help patients manage frustration, social withdrawal, and emotional distress [107]. Psychosocial interventions, including counseling, caregiver training, and support groups, can provide emotional support and practical guidance on navigating the challenges of PPA [106]. Educating families about the disorder’s progression and offering structured social support networks can help mitigate relationship strain, ensuring that patients maintain a sense of belonging and emotional well-being [107].

As PPA progresses, long-term care and palliative support become essential considerations. End-of-life care planning, including legal and financial decision-making, should be discussed early, while the patient can still express their preferences [107]. The involvement of palliative and hospice care teams ensures that both physical symptoms and emotional distress are managed effectively, enhancing the patient’s comfort and dignity [102]. Ethical considerations, particularly in decision-making capacity and patient autonomy, pose challenges as cognitive decline progresses. Healthcare providers must advocate for patient-centered care, ensuring that individuals with PPA retain agency over their treatment decisions for as long as possible while supporting families in making informed, compassionate choices when the patient’s ability to communicate diminishes [94].

### 4.5. Study Strengths and Limitations

Our systematic review provides the most up-to-date and comprehensive synthesis of therapeutic modalities for PPA by integrating more than 50 studies, which is an exceptional volume for such a rare neurodegenerative disorder. Although the inclusion of single-case designs confers reduced internal validity and bias susceptibility, their inclusion is warranted for PPA being a rare neurodegenerative disorder with a limited number of well-conducted RCTs. Omitting lower-level evidence would leave sizeable knowledge gaps and risk of publication bias. Nevertheless, considerable heterogeneity in sample sizes, study designs, intervention protocols, and outcome metrics limits direct comparison across studies and precludes firm subtype-specific recommendations. Standardizing these elements in future research will be essential for generating more uniform and clinically actionable conclusions.

### 4.6. Future Directions

Given the high heterogeneity observed in this review, future research should focus on standardizing intervention protocols and outcome measures to improve comparability between studies. Conducting large-scale, multicenter RCTs with well-defined diagnostic criteria will be essential to establish evidence-based treatment guidelines.

## 5. Conclusions

Our review demonstrates that several therapeutic interventions are effective in improving language function in individuals with PPA. Speech-language therapy, particularly when combined tDCS, has been associated with consistent gains in naming ability, speech fluency, and lexical retrieval. Group-based therapy approaches have also shown beneficial effects in enhancing both communication skills and social interaction. Conversely, NIBS, when used alone, has shown limited effectiveness, and pharmacological interventions have not yielded substantial improvements in language function. Importantly, across all included studies, the reviewed interventions were reported to be safe and well-tolerated, with no significant adverse effects. These findings support the adoption of personalized, multimodal therapeutic strategies that integrate behavioral and neuro-modulatory interventions to optimize communication outcomes in individuals with PPA.

## Figures and Tables

**Figure 1 jcm-14-03063-f001:**
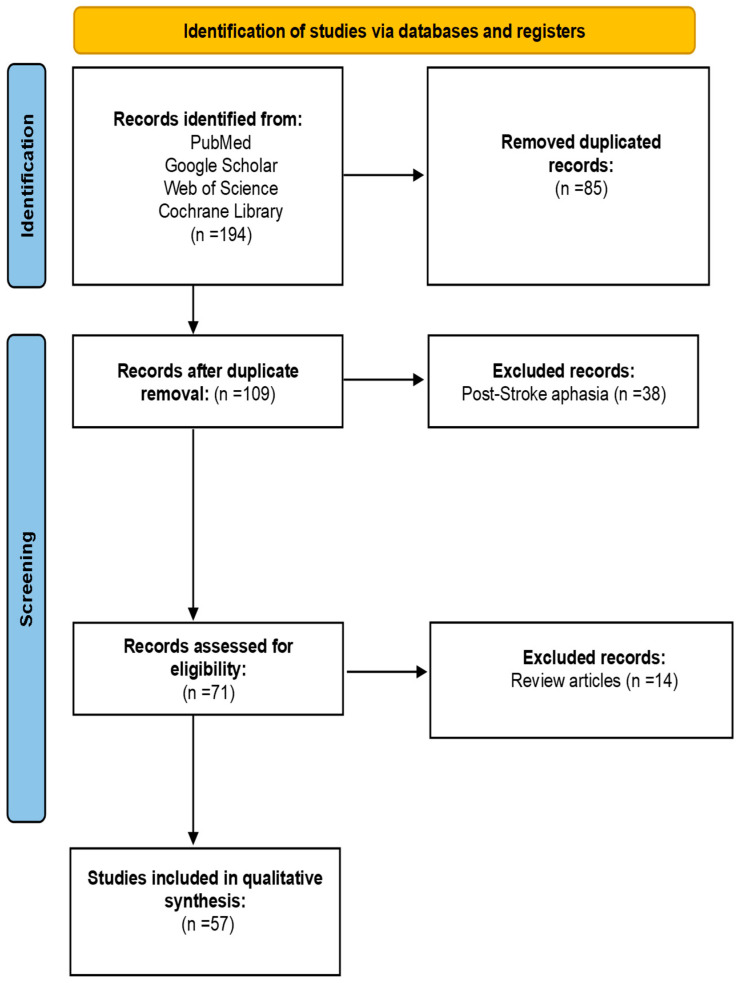
Prisma flow diagram outlining literature review and study selection process.

**Table 1 jcm-14-03063-t001:** Study Characteristics.

Study	Publication Year	Country	Study Design	Population	PPA Subtype	Therapy Used	Treatment Duration	Age, Mean (SD) or Range	Gender (M/F)
Tsapkinia [28]	2018	USA	Randomized controlled trial	36	nfvPPA: 14, svPPA: 10, lvPPA: 12	tDCS and language/spelling therapy	6 w	N/A	20/16
Rogalski [29]	2022	USA	Prospective experimental study	49	nfvPPA: 15, svPPA: 4, lvPPA: 18, other variants: 12	Speech-language therapy	N/A	67.1 (7.3)	25/24
Joubert [30]	2023	Canada	Single case experimental study	1	svPPA: 1	Smartphone-based cognitive and picture naming therapy	5 W	66	0/1
Meyer [31]	2023	USA	Prospective experimental study	37	N/A	Phonological and orthographic treatment	N/A	70.61 (7.49)	N/A
Aguiar [32]	2020	USA	Randomized controlled trial	40	nfvPPA: 15, lvPPA: 17, svPPA: 8	Picture naming, spelling and tDCS	3 W	67.68 (6.76)	22/18
Nissim [33]	2022	USA	Prospective experimental study	12	lvPPA: 8, svPPA: 2, nfvPPA: 2	HD-tDCS and CILT	4 W	66.92 (6.37)	8/4
Henry [34]	2018	USA	Prospective experimental study	10	nfvPPA: 10	VISTA	N/A	67.70 (5.5)	4/6
Pytel [35]	2021	Spain	Randomized controlled trial	20	nfvPPA: 16, svPPA: 4	rTMS	10 w	N/A	8/12
Montagut [36]	2021	Spain	Prospective experimental study	8	svPPA: 8	Errorless learning therapy	8 w	64 (10.5)	4/4
Huang [37]	2023	China	Randomized controlled trial	40	nfvPPA: 16, svPPA: 12, lvPPA: 12 (20 patients in control, and 20 patients in intervention)	rTMS	4 w	65.2 (6.6)	19/21
Borrego-Ecija [38]	2023	Spain	Randomized controlled trial	15	svPPA: 4, lvPPA: 5, nfvPPA: 6	tDCS and speech therapy	4 w	63 (8.4)	5/10
Schaffer [39]	2022	USA	Single case experimental study	1	nfvPPA: 1	VISTA	6 w	72	1/0
Schaffer [40]	2021	USA	Single case experimental study	1	nfvPPA: 1	Spontaneous speech and fluency therapy	6 w	78	0/1
Meyer [41]	2018	USA	Prospective experimental study	14	svPPA: 5, lvPPA: 9	Phonological and orthographic treatment	24 w	65.6 (5.3)	5/9
Meyer [42]	2019	USA	Prospective experimental study	26	svPPA: 5, lvPPA: 9, nfvPPA: 12	Phonological and orthographic treatment	24 w	69.2 (8.1)	12/14
Farrajota [43]	2012	Portugal	Prospective experimental study	20	Intervention: nfvPPA: 2, svPPA: 2, lvPPA: 6 Control: nfvPPA: 0, svPPA: 6, lvPPA: 4	SLT	N/A	68 (7.8)	14/6
Cotelli [44]	2014	Italy	Randomized controlled trial	16	lvPPA: 16	tDCS and ICAT	2 w	66.9 (8.2)	6/10
Dial [45]	2019	USA	Non-randomized trial	31	nfvPPA: 10, svPPA: 10, lvPPA: 11	LRT and VISTA	N/A	67.6 (3.9)	13/18
Krajenbrink [46]	2020	Australia	Single case experimental study	1	svPPA: 1	RRIPP and COEN	6 w	60	1/0
Tylor-Rubin [47]	2022	Australia	Case series	4	svPPA: 3, lvPPA: 1	RRIPP	6 w	69.25 (5.2)	0/4
Gervits [48]	2016	USA	Prospective experimental study	6	lvPPA: 4, nfvPPA: 2	tDCS	2 w	66.2 (5.7)	1/5
Rogalski [17]	2016	USA	Prospective experimental study	31	N/A	SLT	8 w	67.2 (6.9)	13/18
Jokel [49]	2016	Canada	Case Series	4	svPPA: 4	Errorless learning therapy	10 w	61.25 (7.75)	N/A
Meyer [50]	2015	USA	Single case experimental study	1	lvPPA: 1	Phonological and orthographic treatment	48 w	69	0/1
Trebbastoni [51]	2013	Italy	Single case experimental study	1	lvPPA: 1	hf-rTMS	20 s	50	1/0
Meyer [52]	2016	USA	Case series	3	nfvPPA: 1, svPPA: 1, lvPPA: 1	Phonological and orthographic treatment	24 w	61.6	1/2
Shah-Basak [53]	2022	Canada	Single case experimental study	1	nfvPPA: 1	Written naming and HD-tDCS	2 w	67	1/0
Hung [54]	2017	USA	Prospective experimental study	4	svPPA: 3, lvPPA: 1	Semantic feature training and tDCS	2 w	66.6 (8.56)	3/2
Themistocleous [55]	2021	USA	Prospective experimental study	8	nfvPPA with AOS: 8	tDCS and speech therapy	6 w	66 (8.3)	4/4
Jafari [56]	2018	Iran	Single case experimental study	1	nfvPPA with AOS: 1	Cueing hierarchy and story-retelling therapy	8 w	56	0/1
Aguiar [56]	2022	USA	Single case experimental study	1	lvPPA: 1	LeFT and anodal tDCS	12 w	72	1/0
Lavoie [58]	2020	Canada	Case series	5	svPPA: 2, lvPPA: 3	Tablet-based SFA	4 w	72	2/3
Croot [59]	2015	Australia	Case series	2	lvPPA: 2	RRIPP	2 w	67 (13)	1/1
Tsapkini [60]	2014	USA	Crossover trial	6	nfvPPA: 2, lvPPA: 4	tDCS	N/A	N/A	3/3
Fenner [61]	2019	USA	Randomized controlled trial	11	nfvPPA: 6, lvPPA: 5	tDCS and verb naming and spelling therapy	N/A	67.6 (7.7)	5/6
Henry [62]	2019	USA	Prospective experimental study	18	svPPA: 9, lvPPA: 9	LRT	12 w	65.28 (8.32)	7/11
Henry [63]	2013	USA	Case series	2	svPPA: 1, lvPPA: 1	Structured oral reading therapy	12 w	57 (3)	0/1
Kim [64]	2017	USA	Case series	2	lvPPA: 2	LRC	N/A	68.5 (5.5)	N/A
Dressel [65]	2010	Germany	Single case experimental study	1	svPPA: 1	Naming therapy and cueing hierarchies	4 w	48	1/0
Jokel [66]	2010	Canada	Single case experimental study	1	svPPA: 1	Errorless learning therapy	12 w	N/A	1/0
Senaha [67]	2010	Brazil	Single case experimental study	3	svPPA: 3	Lexical reacquisition (errorless learning)	24–72 w	62.66 (10.14)	2/1
Beeson [68]	2011	USA	Single case experimental study	1	lvPPA: 1	Semantic retrieval therapy	2 w	77	1/0
Mayberry [69]	2011	UK	Prospective experimental study	2	svPPA: 2	Relearning therapy	3 w	58.5 (6.5)	1/1
Jokel [70]	2012	Canada	Prospective experimental study	7	svPPA: 7	Errorless learning therapy	8–12 w	68.28 (9.96)	3/4
Henry [71]	2013	USA	Single case experimental study	1	lvPPA: 1	LRC	4 w	73	2/0
Hoffman [72]	2015	UK	Case series	3	svPPA: 3	Vocabulary relearning	3 w	59.57 (2.87)	0/3
Suárez-González [73]	2015	Spain, UK, Australia	Single case experimental study	1	svPPA: 1	COEN and naming Therapy	12 w	57	0/1
Roncero [74]	2017	Canada	Randomized controlled trial	10	nfvPPA: 6, svPPA: 2, lvPPA: 2	tDCS and object naming	3 w	67.4 (5.9)	7/3
Ficek [75]	2018	USA	Randomized controlled trial	36	Intervention: nfvPPA: 5, lvPPA: 9, svPPA: 3, Control: nfvPPA: 8, lvPPA: 6, svPPA: 5	tDCS and language therapy	6 w	67.2 (6.5)	13/11
Grasso [76]	2019	USA	Single case experimental study	2	Mixed PPA: 1, Mild cognitive Impairment: 1	LRT	N/A	72.5 (6.5)	1/1
Meyer [77]	2017	USA	Case series	21	lvPPA: 9, svPPA: 5, nfvPPA: 7	Orthographic treatment	24 w	69.1 (2.3)	10/11
Suárez-González [78]	2011	UK	Single case experimental study	1	svPPA: 1	COEN and naming therapy	1 w	62	0/1
Croot [79]	2019	Australia	Case series	8	nfvPPA: 3, lvPPA: 2, svPPA: 2, Mixed PPA: 1	LRT	2–4 w	64.87 (5.5)	3/5
Cotelli [80]	2016	Italy	Case series	18	nfvPPA: 18	tDCS	N/A	66.5 (9.5)	9/9
Sayadnasiri [81]	2021	Iran	Single case experimental study	13	nfvPPA: 13	Zolpidem	6 w	58.5 (4.5)	8/5
Wang [82]	2023	USA	Randomized controlled trial	36	lvPPA: 14, nfvPPA: 13, svPPA: 9	tDCS and lexical/semantic therapy	8 w	50–80	19/17
Lerman [83]	2023	Israel	Single case experimental study	1	lvPPA: 1	VNeST	10 w	70	0/1

PPA: primary progressive aphasia; nfvPPA: nonfluent/agrammatic variant primary progressive aphasia; svPPA: semantic variant; lvPPA: logopenic variation of primary progressive aphasia; tDCS: transcranial direct current stimulation; HD-tDCS: high-definition transcranial direct current stimulation; rTMS: repetitive transcranial magnetic stimulation; hf-rTMS: high-frequency repetitive transcranial magnetic stimulation; SLT: speech-language therapy; CILT: constraint-induced language therapy; ICAT: intensive comprehensive aphasia therapy; VISTA: video-implemented script training for aphasia; LRC: lexical retrieval cascade; LRT: lexical retrieval therapy; RRIPP: repetitive rehabilitation for impaired phonological processing; COEN: cognitive and executive naming; LeFT: lexical feature training; AOS: apraxia of speech; SFA: semantic feature analysis; VNeST: verb network strengthening treatment; N/A: not available or not applicable; M/F: male/female; SD: standard deviation.

**Table 2 jcm-14-03063-t002:** Risk of bias assessment for RCTs and crossover trials (Cochrane RoB 2 Tool).

First Author	Year	Selection Bias	Performance Bias	Detection Bias	Attrition Bias	Reporting Bias	Overall Risk
Tsapkinia [28]	2018	Low	Some Concerns	Low	Low	Low	Low
Aguiar [32]	2020	Low	Low	Low	Some Concerns	Low	Some Concerns
Cotelli [44]	2014	Low	Low	Some Concerns	Low	Low	Low
Huang [37]	2023	Low	Low	Low	Low	Some Concerns	Low
Borrego-Ecija [38]	2023	Some Concerns	Some Concerns	Low	Low	Low	Some Concerns
Pytel [37]	2021	Low	Some Concerns	Low	Some Concerns	Low	Some Concerns
Roncero [74]	2017	Low	Low	Low	Low	Low	Low
Ficek [75]	2018	Low	Low	Low	Low	Some Concerns	Low
Wang [82]	2023	Low	Some Concerns	Low	Low	Low	Some Concerns

RoB 2: Cochrane risk of bias tool 2; RCTs: randomized controlled trials.

**Table 3 jcm-14-03063-t003:** Risk of bias assessment for non-randomized and experimental studies (ROBINS-I Tool).

First Author	Year	Bias Due to Confounding	Selection Bias	Classification Bias	Deviations from Intended Interventions	Missing Data	Measurement Bias	Reporting Bias	Overall Risk
Rogalski [29]	2022	Moderate	Moderate	Low	Some Concerns	Low	Low	Some Concerns	Moderate
Meyer [31]	2023	Moderate	Low	Some Concerns	Some Concerns	Low	Low	Some Concerns	Moderate
Nissim [33]	2022	Moderate	Some Concerns	Low	Low	Some Concerns	Low	Some Concerns	Moderate
Henry [34]	2018	Moderate	Low	Low	Some Concerns	Low	Low	Some Concerns	Moderate
Montagut [36]	2021	High	Some Concerns	Low	Some Concerns	Some Concerns	Low	High	High
Henry [62]	2019	Moderate	Low	Some Concerns	Some Concerns	Low	Low	Some Concerns	Moderate
Themistocleous [55]	2021	Moderate	Low	Low	Some Concerns	Low	Low	Low	Moderate
Sayadnasiri [81]	2021	High	Some Concerns	Low	Some Concerns	Low	Low	High	High
Lerman [83]	2023	Moderate	Low	Some Concerns	Low	Low	Low	Some Concerns	Moderate

ROBINS-I: Risk of bias in non-randomized studies of interventions.

**Table 4 jcm-14-03063-t004:** Risk of bias assessment for single-case designs and case series (JBI Checklist).

First Author	Year	Patient Selection	Intervention Clearly Described	Outcome Measures Reliable	Follow-Up Adequate	Ethical Considerations	Overall Quality
Joubert [30]	2023	Yes	Yes	Yes	No	Yes	Moderate
Schaffer [39]	2022	Yes	Yes	No	No	Yes	Low
Schaffer [40]	2021	Yes	Yes	Yes	No	Yes	Moderate
Meyer [41]	2018	Yes	Yes	Yes	No	Yes	Moderate
Meyer [42]	2019	Yes	Yes	Yes	No	Yes	Moderate
Farrajota [43]	2012	No	Yes	Yes	No	Yes	Low
Dial [45]	2019	Yes	Yes	Yes	No	Yes	Moderate
Krajenbrink [46]	2020	No	Yes	No	No	Yes	Low
Taylor-Rubin [47]	2022	Yes	Yes	Yes	No	Yes	Moderate
Gervits [48]	2016	No	Yes	No	No	Yes	Low
Rogalski [17]	2016	No	Yes	No	No	Yes	Low
Jokel [49]	2016	Yes	Yes	Yes	No	Yes	Moderate
Meyer [50]	2015	No	Yes	No	No	Yes	Low
Trebbastoni [51]	2013	No	Yes	No	No	Yes	Low
Meyer [52]	2016	No	Yes	No	No	Yes	Low
Shah-Basak [53]	2022	Yes	Yes	Yes	No	Yes	Moderate
Hung [54]	2017	Yes	Yes	Yes	No	Yes	Moderate
Beeson [68]	2011	Yes	Yes	Yes	No	Yes	Moderate
Henry [71]	2013	No	Yes	No	No	Yes	Low
Hoffman [72]	2015	No	Yes	No	No	Yes	Low
Suárez-González [73]	2015	No	Yes	No	No	Yes	Low
Roncero [74]	2017	Yes	Yes	Yes	No	Yes	Moderate

JBI: Joanna Briggs Institute.

**Table 5 jcm-14-03063-t005:** Summary of implemented intervention and their outcomes in PPA.

Outcomes	Therapy Used	Authors	PPA Subtype	Follow-Up	Results (Intervention vs. Control)
Letter (naming) accuracy during spelling of trained words	tDCS and language/spelling therapy	Tsapkinia 2018 [28]	nfvPPA: 14, svPPA: 10, lvPPA: 12	Post-Treatment	Intervention group language gains were improved and maintained significantly
				2 weeks	
				2 months	
		Ficek 2018 [75]	Intervention: nfvPPA: 5, lvPPA: 9, svPPA: 3 Control: nfvPPA: 8, lvPPA: 6, svPPA: 5	Post-Treatment	
	tDCS and picture naming and spelling therapy	Aguiar 2022 [57]	lvPPA: 1	Post-Treatment	
				2-weeks	
				2-months	
	tDCS and lexical/semantic therapy	Wang 2023 [82]	lvPPA: 14, nfvPPA: 13, svPPA: 9	Post-treatment	
				2 weeks	
				2 months	
	Phonological and orthographic treatment	Meyer 2018 [41]	svPPA: 5, lvPPA: 9	Post-Treatment	
	Errorless learning therapy	Jokel 2012 [70]	svPPA: 7	Post-Treatment	
Letter (naming) accuracy during spelling of untrained words	tDCS and language/spelling therapy	Tsapkinia 2018 [28]	nfvPPA: 14, svPPA: 10, lvPPA: 12	Post-Treatment	
				2 weeks	
				2 Months	
	tDCS and picture naming and spelling	Aguiar 2022 [57]	lvPPA: 1	Post-Treatment	
				2-weeks	
				2-months	
Reading accuracy	rTMS	Pytel 2021 [35]	nfvPPA: 16, svPPA: 4	Post-Treatment	
	tDCS and speech therapy	Borrego-Ecija 2023 [38]	svPPA: 4, lvPPA: 5, nfvPPA: 6	Post-Treatment	
	Structured oral reading therapy	Henry 2013 [71]	lvPPA: 1	Post-Treatment	Reading accuracy improved after therapy implementation
				3-months	
				6-months	
				1-year	
Production of correct, intelligible scripted words (Speech)	VISTA	Henry 2018 [34]	nfvPPA: 10	Post-Treatment	Substantial improvement and maintenance in production of scripted word
				3-months	
				6-months	
				1-year	
	VISTA	Schaffer 2022 [39]	nfvPPA: 1	Post-Treatment	
				3-months	
				6-months	
				1-year	
Spontaneous speech and fluency	Smartphone-based cognitive and picture naming	Joubert 2023 [30]	svPPA: 1	Post-Treatment	Intervention group showed positive reinforcement of spontaneous speech and fluency with different generalization and maintenance reinforcements
				6-months	
	HD-tDCS and CILT	Nissim 2022 [33]	lvPPA: 8, svPPA: 2, nfvPPA: 2	Post-Treatment	
				6-weeks	
	tDCS and speech therapy	Borrego-Ecija 2023 [38]	svPPA: 4, lvPPA: 5, nfvPPA: 6	Post-Treatment	
	Semantic feature training and tDCS	Hung 2017 [54]	svPPA: 3, lvPPA: 1	Post-Treatment	
				6-months	
	tDCS and speech therapy	Themistocleous 2021 [55]	nfvPPA with AOS: 8	Post-Treatment	
				2-months	
	tDCS	Gervits 2016 [48]	lvPPA: 4, nfvPPA: 2	2-weeks	
				6-weeks	
				12-weeks	
	LeFT and anodal tDCS	Aguiar 2022 [57]	lvPPA: 1	Post-Treatment	
				2-weeks	
				2-months	
	tDCS	Tsapkini 2014 [60]	nfvPPA: 2, lvPPA: 4	Post-Treatment	
				2-weeks	
				2-months	
	rTMS	Pytel 2021 [36]	nfvPPA: 16, svPPA: 4	Post-Treatment	
	hf-rTMS	Trebbastoni 2013 [51]	lvPPA: 1	Post-Treatment	
	VISTA	Henry 2018 [34]	nfvPPA: 10	1-year	
	VISTA	Schaffer 2022 [39]	nfvPPA: 1	Post-Treatment	
	Spontaneous speech and fluency	Schaffer 2021 [40]	nfvPPA: 1	Post-Treatment	
				3-months	
				6-months	
				1-year	
	Errorless learning therapy	Jokel 2016 [49]	svPPA: 4	Post-Treatment	
	Phonological and orthographic treatment	Meyer 2016 [41]	svPPA: 5, lvPPA: 9	1-month	
	Cueing hierarchy and story-retelling therapy	Jafari 2018 [56]	nfvPPA with AOS: 1	Post-Treatment	
	RRIPP	Croot 2015 [59]	lvPPA: 2	Patient 1: Post-Treatment, 1-month	
				Patient 2: Post-Treatment, 9-months	
	Structured oral reading therapy	Henry 2013 [71]	lvPPA: 1	Post-Treatment	
				3-months	
				6-months	
				1-year	
	LRC	Kim 2017 [64]	lvPPA: 2	Post-Treatment	
				2-months	
AOS	VISTA	Henry 2018 [34]	nfvPPA: 10	Post-Treatment	Intervention group showed significant improvement in speech accuracy that was maintained post-treatment
				3-months	
				6-months	
				1-year	
		Schaffer 2022 [39]	nfvPPA: 1	Post-Treatment	
				3-months	
				6-months	
				1-year	
		Dial 2019 [45]	nfvPPA: 10, svPPA: 10, lvPPA: 11	Post-Treatment	
				3-months	
				6-months	
				1-year	
	Structured oral reading therapy	Henry 2013 [71]	lvPPA: 1	Post-Treatment	
				3-months	
				6-months	
				1-year	
		Meyer 2016 [52]	nfvPPA: 1, svPPA: 1, lvPPA: 1	1-month	
	LeFT and tDCS	Aguiar 2022 [57]	lvPPA: 1	Post-Treatment	
				2-weeks	
				2-months	
	tDCS+ speech therapy	Themistocleous 2021 [55]	nfvPPA with AOS: 8	Post-Treatment	
				2-months	
	rTMS	Pytal 2021 [35]	nfvPPA: 16, svPPA 4	Post-Treatment	
	HD-tDCS and story retelling with cueing hierarchy	Jafari 2018 [56]	nfvPPA with AOS: 1	Post-Treatment	
		Shah-Basak 2022 [53]	nfvPPA: 1	5-days	
				3-months	
Word retrieval	tDCS and language therapy	Tsapkinia 2018 [28]	nfvPPA: 14, svPPA: 10, lvPPA: 12	Post-Treatment	Intervention group showed significant improvement in word retrieval potential that was maintained post-treatment
				2-weeks	
				2-months	
		Cotelli 2014 [44]	lvPPA: 16	2-weeks	
				12-weeks	
		Roncero 2017 [74]	nfvPPA: 6, svPPA: 2, lvPPA: 2	2-weeks	
		Ficek 2018 [75]	Intervention:nfvPPA: 5, lvPPA: 9, svPPA: 3 Control: nfvPPA: 8, lvPPA: 6, svPPA: 5	Post-Treatment	
		Huang 2023 [37]	nfvPPA: 16, svPPA: 12, lvPPA: 12 (20 patients in control, and 20 patients in intervention)	1-month	
				3-months	
				6-months	
		Borrego-Ecija 2023 [38]	svPPA: 4, lvPPA: 5, nfvPPA: 6	Post-Treatment	
		Tsapkini 2014 [60]	nfvPPA: 2, lvPPA: 4	Post-Treatment	
				2-weeks	
				2-months	
		Gervits 2016 [47]	lvPPA: 4, nfvPPA: 2	2-weeks	
				6-weeks	
				12-weeks	
		Lerman 2023 [83]	lvPPA: 1	Post-Treatment	
		Wang 2023 [82]	lvPPA: 14, nfvPPA: 13, svPPA: 9	Post-Treatment	
				2-weeks	
				2-months	
	SLT	Rogalski 2022 [30]	nfvPPA: 15, svPPA: 4, lvPPA: 18, other variants: 12	2-months	
				6-months	
		Farrajota 2012 [43]	Intervention: nfvPPA: 2, svPPA: 2, lvPPA: 6 Control: nfvPPA: 0, svPPA: 6, lvPPA: 4	Post-Treatment	
		Henry 2019 [62]	svPPA: 9, lvPPA: 9	Post-Treatment	
				3-months	
				6-months	
				1-year	
		Jokel 2016 [49]	svPPA: 4	Post-Treatment	
		Meyer 2017 [77]	lvPPA: 9, svPPA: 5, nfvPPA: 7	1-month	
	tDCS and spelling/picture naming	Aguiar 2020 [32]	nfvPPA: 15, lvPPA: 17, svPPA: 8	Post-Treatment	
				2-weeks	
				2-months	
		Croot 2019 [79]	nfvPPA: 3, lvPPA: 2, svPPA: 2, Mixed PPA: 1	Post-Treatment	
		Borrego-Ecija 2023 [38]	svPPA: 4, lvPPA: 5, nfvPPA: 6	Post-Treatment	
	Phonological and orthographic treatment	Meyer 2018 [41]	svPPA: 5, lvPPA: 9	Post-Treatment	
		Meyer 2019 [42]	svPPA: 5, lvPPA: 9, nfvPPA: 12	1-month	
		Meyer 2015 [50]	lvPPA: 1	1-week	
				8-months	
				1-year	
				20-months	
				3-years	
		Huang 2023 [37]	nfvPPA: 16, svPPA: 12, lvPPA: 12 (20 patients in control, and 20 patients in intervention)	1-month	
				3-months	
				6-months	
	Errorless learning therapy	Jokel 2010 [66]	svPPA: 1	Post-Treatment	
		Montagut 2021 [36]	svPPA: 8	Post-Treatment	
				1-month	
				3-months	
				6-months	
		Gervits 2016 [48]	lvPPA: 4, nfvPPA: 2	2-weeks	
				6-weeks	
				12-weeks	
	LRC	Beeson 2011 [68]	lvPPA: 1	Post-Treatment	
				3-weeks	
				4-months	
				6-months	
		Henry 2013 [63]	lvPPA: 1, svPPA: 1	Post-Treatment	
				3-months	
				6-months	
				1-year	
		Meyer 2015 [50]	lvPPA: 1	1-week	
				8-months	
				1-year	
				20-months	
				3-years	
		Grasso 2019 [70]	svPPA: 7	Post-Treatment	
				3-months	
				6-months	
		Henry 2018 [34]	nfvPPA: 10	Post-Treatment	
				3-months	
				6-months	
				1-year	
	tDCS and CILT	Nissim 2022 [33]	lvPPA: 8, svPPA: 2, nfvPPA: 2	Post-Treatment	
				6-weeks	
	Semantic feature training and tDCS	Hung 2017 [54]	svPPA: 3, lvPPA: 1	Post-Treatment	
				6-months	
		Lavoie 2020 [58]	svPPA: 2, lvPPA: 3	Post-Treatment	
	RRIPP and COEN	Krajenbrink 2020 [46]	svPPA: 1	Post-Treatment	
				2-weeks	
		Tylor-Rubin 2022 [47]	svPPA: 3, lvPPA: 1	Post-Treatment	
	VISTA	Henry 2018 [34]	nfvPPA: 10	Post-Treatment	
				3-months	
				6-months	
				1-year	
		Henry 2019 [66]	svPPA: 9, lvPPA: 9	Post-Treatment	
				3-months	
				6-months	
				1-year	
		Schaffer 2022 [39]	nfvPPA: 1	Post-Treatment	
				3-months	
				6-months	
				1-year	
	Smartphone-based cognitive and picture naming	Joubert 2023 [30]	svPPA: 1	Post-Treatment	
				6-months	
	rTMS	Pytel 2021 [36]	nfvPPA: 16, svPPA: 4	Post-Treatment	
Reading abilities	Phonological and orthographic treatment	Meyer 2019 [42]	svPPA: 5, lvPPA: 9, nfvPPA: 12	1-month	Intervention group showed significant improvement in reading accuracy that was maintained post-treatment at various intervals
		Meyer 2018 [41]	svPPA: 5, lvPPA: 9	Post-Treatment	
		Henry 2013 [63]	svPPA: 1, lvPPA: 1	Post-Treatment	
				3-months	
				6-months	
				1-year	
	Errorless learning therapy	Jokel 2010 [66]	svPPA: 1	Post-Treatment	
		Senaha 2010 [67]	svPPA: 3	Post-Treatment	
		Montagut 2021 [36]	svPPA: 8	Post-Treatment	
				1-month	
				3-months	
				6-months	
	rTMS	Pytel 2021 [35]	nfvPPA: 16, svPPA: 4	Post-Treatment	
		Henry 2019 [62]	svPPA: 9, lvPPA: 9	Post-Treatment	
				3-months	
				6-months	
				1-year	
Repetition abilities	tDCS and language therapy	Tsapkinia 2018 [28]	nfvPPA: 14, svPPA: 10, lvPPA: 12	Post-Treatment	Intervention group showed improvement in reading accuracy that was maintained post-treatment at various intervals, some studies reported mixed results
				2-weeks	
				2-months	
		Cotelli 2016 [80]		Post-Treatment	
				3-months	
		Ficek 2018 [75]	Intervention: nfvPPA: 5, lvPPA: 9, svPPA: 3 Control: nfvPPA: 8, lvPPA: 6, svPPA: 5	Post-Treatment	
		Gervits 2016 [48]	lvPPA: 4, nfvPPA: 2	2-weeks	
				6-weeks	
				12-weeks	
		Roncero 2017 [74]	nfvPPA: 6, svPPA: 2, lvPPA: 2	2-weeks	
		Fenner 2019 [61]	nfvPPA: 6, lvPPA: 5	Post-Treatment	
				2-weeks	
				2-months	
	SLT	Farrajota 2012 [43]	Intervention: nfvPPA: 2, svPPA: 2, lvPPA: 6 Control: nfvPPA: 0, svPPA: 6, lvPPA: 4	Post-Treatment	
		Meyer 2019 [42]	svPPA: 5, lvPPA: 9, nfvPPA: 12	1-month	
		Rogalski 2022 [29]	nfvPPA: 15, svPPA: 4, lvPPA: 18, other variants: 12	2-months	
				6-months	
	Errorless learning therapy	Jokel 2010 [66]	svPPA: 1	Post-Treatment	
		Jokel 2016 [49]	svPPA: 4	Post-Treatment	
		Themistocleous 2021 [55]	nfvPPA with AOS: 8	Post-Treatment	
				2-months	
	Cognitive and picture naming therapy	Joubert 2023 [30]	svPPA: 1	Post-Treatment	
				6-months	
	Phonological and orthographic therapy	Meyer 2015 [50]	lvPPA: 1	1-week	
				8-months	
				1-year	
				20-months	
				3-years	
		Meyer 2016 [41]	svPPA: 5, lvPPA: 9	1-month	
		Meyer 2019 [42]	svPPA: 5, lvPPA: 9, nfvPPA: 12	1-month	
	tDCS and cognitive therapy (ICAT)	Cotelli 2014 [44]	lvPPA: 16	2-weeks	
				12-weeks	
		Tsapkini 2014 [60]	nfvPPA: 2, lvPPA: 4	Post-Treatment	
				2-weeks	
				2-months	
	rTMS and language therapy	Pytel 2021 [35]	nfvPPA: 16, svPPA: 4	Post-Treatment	
	VISTA	Henry 2018 [34]	nfvPPA: 10	Post-treatment	
				3-months	
				6-months	
				1-year	
	LRC	Grasso 2019 [76]	Mixed PPA: 1, Mild cognitive Impairment: 1	Post-Treatment	
				3-months	
				6-months	
		Suárez-González 2015 [73]	svPPA: 1	Post-Treatment	
		Croot 2019 [79]	nfvPPA: 3, lvPPA: 2, svPPA: 2, Mixed PPA: 1	Post-Treatment	

PPA: primary progressive aphasia; nfvPPA: nonfluent/agrammatic variant primary progressive aphasia; svPPA: svPPA: semantic variant primary progressive aphasia; lvPPA: logopenic variant of primary progressive aphasia; AOS: apraxia of speech; CILT: constraint-induced language therapy; COEN: cognitive and executive naming; HD-tDCS: high-definition transcranial direct current stimulation; hf-rTMS: high-frequency repetitive transcranial magnetic stimulation; ICAT: intensive comprehensive aphasia therapy; LeFT: lexical feature training; LRC: lexical retrieval cascade; RRIPP: repetitive rehabilitation for impaired phonological processing; rTMS: repetitive transcranial magnetic stimulation; SLT: speech-language therapy; tDCS: transcranial direct current stimulation; VISTA: video-implemented script training for aphasia.

## Data Availability

All data generated or analyzed during this study are included in this published article [and its Appendix A].

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
