# Peer review of "Safety and Efficacy of Different Therapeutic Interventions for Primary Progressive Aphasia: A Systematic Review"

_jcm, 2025, doi:10.3390/jcm14093063_

Round 1
Reviewer 1 Report
Comments and Suggestions for Authors
Dear Authors,
the chosen issue is very relevant and interesting as there is no effective treatment for neurodegeneration disease with cognitive and speech impairment. Identification of therapeutic strategics is crucial for patients and their caregivers. However, this manuscript needs a major revision in many aspects.
Firstly, language must be improved. There are many misprints, typos, punctuation mistakes, repetitions and illogical sentences in abstract and in macusrpit. Additionally, English correction is required.
Some chosen keywords are not qualified as MeSH, for example: term I is not a MeSH, you should use Aphasia, Primary Progressive, the same with the term Neurodegenerative – Neurodegenerative Diseasesinstead. The term Speech-Language Theraphy is aso not a MeSH. It should be corrected.
The Introduction provides sufficient background about the characteristics and subtypes of PPA, however there is a lack of pathophysiology as well as therapeutic procedures. Please check also the used references, for example the reference nr 16 (Bott NT, Radke A, Stephens ML, Kramer JH (2014) Frontotemporal dementia: diagnosis, deficits and management. Neurodegener Dis Manag 4:439–454.
https://doi.org/10.2217/NMT.14.34) is not relevant to diverse neurodegenerative mechanism in FTD and Alzheimer Disease.
The purpose of the review is well defined (assessment of efficacy and safety in different therapeutic interventions for PPA).
In chapter Search Strategy, there are typos and misprint in used MeSHes, whereas chapter Eligibility Criteriacontains a reintroduction of the abbreviations already used in the introduction, the same in Assessment of quality of the included studies concerning RCTs.
The eligibility criteria for this systematic review are too broad including RCTs, experimental studies and case reports. The criteria should be made more precise and limited to, for example, RCTS, but I can understand the aim and idea of this review to show all tested therapeutic strategies. However, RCTs-based outcome assessment is more valuable and points to therapies that can be implemented or developed. I would not include in the assessment the results of case reports both experimental studies. In my opinion this major revision should be implemented before acceptation of the manuscript.
Table 1 demonstrates the characteristics of the enrolled studies. After exclusions of case reports and experimental studies, it would be valuable to add the results of the study (whether the intervention applied resulted in statistically significant benefits).
Table 2 shows the risk of bias, and it brings a lot of value in this review.
Summary of implemented interventions and their outcomes in PPA in Table 3 is also valuable as a demonstration of the effectiveness of specific methods. However, the results of case studies or experimental studies may be misleading.
The manuscript needs a clear separation between results and discussion, as there is a lack of clarity, and it can confuse the reader. It is best to divide the chapters numerically according to 1. Introduction, 2. Methods, 3. Results, 4. Discussion, and 5. Conclusions. Addidiontally subsections should be divided into subchapters for example: 3.1, 3.2 and 3.2.1.
In my opinion, the section Treatment gain should be included in Disccusion. A clear separation between the analysed topics is required, for example the Treatment gain should be clearly separated from maintenance of therapeutic effects and neuroplastic and metabolic changes, as the reader can be confused. Due to too many low-quality studies included in the review, it does not reveal consistent and clear conclusions. The numerous repetitions and the reference to almost a dozen references is unacceptable and illegible as in following paragraphs:
- Naming and word-finding for trained and untrained world
- Spontaneuos speech or fluency
- Auditory verbal comprehension
- Repetition
- Apraxia of speech
- Reading abilities
- Cognitive functions
- Maintenece of Therapeutic Effects
The inclusion of following outcomes: cognitive function, maintenance of the therapeutic effects and neuroplastic and functional reorganisation and metabolic changes is appreciated but needs to be reassessed after exclusion of experimental studies and case reports.
In Discussion, section Summary of Findings there is an unacceptable lack of supporting references. Section Clinical Implictions of PPA and Phatophisiology and Treatment Considerations should be included in Introdcuction, the Discussion should correspond with the aim of this review. The section Burden of PPAincluding Financial Implications, Psychological and Emotional Challenges and Social Imapct and Stigma the manuscript in its present form should not be published are not relevant in this issue.
As authors pointed out, the significant heterogeneity among included studies is one of the greatest limitations of the study. The inclusions criteria of this study should be limited to RCTs or for example to specific subtypes of PPA as the conclusions are misleading and inconsistent.
The Conclusions are not consistent with the aim of this review. Effective or ineffective forms of therapy are not identified, as well as there is nothing about the safety of this interventions whereas the title of this manuscript is Safty and Efficacy of Different Treatment Interevensions in PPA.
A major revision of methodology (including only RTCs), results and discussion, must be implemented before manuscript acceptance.
Sincerely Yours,
Comments on the Quality of English LanguageEnglish correction is required.
Author Response
Comment 1: [Firstly, language must be improved. There are many misprints, typos, punctuation mistakes, repetitions and illogical sentences in abstract and in macusrpit. Additionally, English correction is required.]
Response 1: [Thank you for your insightful feedback. Please find the revised manuscript with the corrected errors throughout.]
Comment 2: [Some chosen keywords are not qualified as MeSH, for example: term I is not a MeSH, you should use Aphasia, Primary Progressive, the same with the term Neurodegenerative – Neurodegenerative Diseasesinstead. The term Speech-Language Theraphy is aso not a MeSH. It should be corrected.]
Response 2: [Thank you for your insightful feedback. Please find the revised manuscript with the corrected keyword on page 2, line 62]
Comment 3: [The Introduction provides sufficient background about the characteristics and subtypes of PPA, however there is a lack of pathophysiology as well as therapeutic procedures. Please check also the used references, for example the reference nr 16 (Bott NT, Radke A, Stephens ML, Kramer JH (2014) Frontotemporal dementia: diagnosis, deficits and management. Neurodegener Dis Manag 4:439–454.https://doi.org/10.2217/NMT.14.34) is not relevant to diverse neurodegenerative mechanism in FTD and Alzheimer Disease.]
Response3: [Thank you for your insightful feedback. The addressed comment can be found on page 4, lines 112–129. Additionally, some general therapeutic approaches were discussed throughout lines 112–158.]
Comment 4: [In chapter Search Strategy, there are typos and misprint in used MeSHes, whereas chapter Eligibility Criteriacontains a reintroduction of the abbreviations already used in the introduction, the same in Assessment of quality of the included studies concerning RCTs.]
Response 4: [Thank you for your insightful feedback. The abbreviations were introduced once throughout the manuscript, and the writing errors were addressed in the Methods section on page 5, lines 175–176]
Comment 5: [The eligibility criteria for this systematic review are too broad including RCTs, experimental studies and case reports. The criteria should be made more precise and limited to, for example, RCTS, but I can understand the aim and idea of this review to show all tested therapeutic strategies. However, RCTs-based outcome assessment is more valuable and points to therapies that can be implemented or developed. I would not include in the assessment the results of case reports both experimental studies. In my opinion this major revision should be implemented before acceptation of the manuscript.]
Response 5: [Thank you for your insightful feedback. We appreciate your recommendation; however, due to the aim of comprehensive coverage and the rarity of the condition—with a limited number of large RCTs—the inclusion of single-case designs is warranted to minimize publication bias. The limitations of single-case designs were discussed on page 40, lines 666–671.]
Comment 6: [Table 1 demonstrates the characteristics of the enrolled studies. After exclusions of case reports and experimental studies, it would be valuable to add the results of the study (whether the intervention applied resulted in statistically significant benefits).]
Response 6: [Thank you for your insightful feedback. We appreciate your recommendation; however, due to the aim of comprehensive coverage and the rarity of the condition—with a limited number of large RCTs—the inclusion of single-case designs is warranted to minimize publication bias. The limitations of single-case designs were discussed on page 40, lines 666–671.]
Comment 7: [Summary of implemented interventions and their outcomes in PPA in Table 3 is also valuable as a demonstration of the effectiveness of specific methods. However, the results of case studies or experimental studies may be misleading.]
Response 7: [Thank you for your insightful feedback. We appreciate your recommendation; however, due to the aim of comprehensive coverage and the rarity of the condition—with a limited number of large RCTs—the inclusion of single-case designs is warranted to minimize publication bias. The limitations of single-case designs were discussed on page 40, lines 666–671.]
Comment 8: [The manuscript needs a clear separation between results and discussion, as there is a lack of clarity, and it can confuse the reader. It is best to divide the chapters numerically according to 1. Introduction, 2. Methods, 3. Results, 4. Discussion, and 5. Conclusions. Addidiontally subsections should be divided into subchapters for example: 3.1, 3.2 and 3.2.1.]
Response 8: [Thank you for your insightful feedback. You can find the addressed points throughout the manuscript]
Comment 9: [In my opinion, the section Treatment gain should be included in Disccusion. A clear separation between the analysed topics is required, for example the Treatment gain should be clearly separated from maintenance of therapeutic effects and neuroplastic and metabolic changes, as the reader can be confused. Due to too many low-quality studies included in the review, it does not reveal consistent and clear conclusions. The numerous repetitions and the reference to almost a dozen references is unacceptable and illegible as in following paragraphs:
- Naming and word-finding for trained and untrained world
- Spontaneuos speech or fluency
- Auditory verbal comprehension
- Repetition
- Apraxia of speech
- Reading abilities
- Cognitive functions
- Maintenece of Therapeutic Effects
The inclusion of following outcomes: cognitive function, maintenance of the therapeutic effects and neuroplastic and functional reorganisation and metabolic changes is appreciated but needs to be reassessed after exclusion of experimental studies and case reports.]
Response 9: [Thank you for your insightful feedback. We have ensured clear separation between all manuscript sections, and the results can be clearly distinguished from the discussion. The reference list is provided according to the inclusion criteria mentioned in the methodology section, with a substantial number of references included. We believe that this comprehensive coverage will offer a thorough assessment and provide a realistic view of the current literature.]
Comment 10: [In Discussion, section Summary of Findings there is an unacceptable lack of supporting references. Section Clinical Implictions of PPA and Phatophisiology and Treatment Considerations should be included in Introdcuction, the Discussion should correspond with the aim of this review. The section Burden of PPAincluding Financial Implications, Psychological and Emotional Challenges and Social Imapct and Stigma the manuscript in its present form should not be published are not relevant in this issue.]
Response 10 : [Thank you for your insightful feedback. Regarding the summary of findings, please find the addressed comments on pages 33–34, lines 457–488. Regarding the clinical implications for the smoothness of flow and the purpose of our review, this section is required to be retained in the discussion section. However, we have included a brief section regarding the pathophysiology and implemented therapeutic approaches in the introduction on pages 4–5, lines 112–158]
Comment 11: [As authors pointed out, the significant heterogeneity among included studies is one of the greatest limitations of the study. The inclusions criteria of this study should be limited to RCTs or for example to specific subtypes of PPA as the conclusions are misleading and inconsistent.]
Response 11: [Thank you for your insightful feedback. We appreciate your recommendation; however, due to the aim of comprehensive coverage and the rarity of the condition—with a limited number of large RCTs—the inclusion of single-case designs is warranted to minimize publication bias. The limitations of single-case designs were discussed on page 40, lines 666–671]
Comment 12: [The Conclusions are not consistent with the aim of this review. Effective or ineffective forms of therapy are not identified, as well as there is nothing about the safety of this interventions whereas the title of this manuscript is Safty and Efficacy of Different Treatment Interevensions in PPA.]
Response 12: [Thank you for your insightful feedback. Please find the addressed comments on page 40, lines 679–691. Safety has been discussed according to the findings provided from the retrieved literature on page 33, lines 428–433.]
Reviewer 2 Report
Comments and Suggestions for Authors
Abstract – Results Section
Correct: “Fifty-seven studies with 655patients were included…”
Abstract – Conclusion Section
Original: “and lack of standardized measurement methos drawing a firm conclusion is difficult.”
Introduction Section
Remove the first sentence due to redundancy:
“Frontotemporal dementias are neurodegenerative illnesses that begin with symptoms of frontal and/or temporal lobe dysfunction.”
Outcome Measures Section
Please clarify the outcome measures by naming the specific scores used in the included studies. For example, if available, indicate whether studies used standardized tools such as the Western Aphasia Battery or the Boston Naming Test. If these are unavailable, at least specify the type of data extracted, e.g., language performance metrics, qualitative descriptions, or pre/post therapy changes in communication.
Search Results Section
Correct final sentence:
“Ultimately, 57 studies involving 655patients met the inclusion criteria and were included in the review.”
Study Characteristics Section
Please expand on how the term “experimental study” is defined by the authors. For example: Did it include any study involving an intervention with pre/post measures regardless of controls?. Were case series with intervention considered experimental, even if lacking a comparator group?. Clarify whether pilot trials, feasibility studies, or single-case experimental designs were classified under this term.
Discussion Section
It is crucial to clearly explain why this systematic review includes single-case designs, case reports, and case series, instead of limiting inclusion to controlled trials such as the 2025 meta-analysis by Lomi, which did not find evidence supporting the efficacy of speech and language therapy.
You should justify this methodological choice by addressing:
The limited number of RCTs in the field of frontotemporal dementia and speech therapy.
The exploratory or hypothesis-generating value of these lower-evidence studies.
The individualized nature of interventions in neurodegenerative diseases, which may lend themselves to single-case approaches.
However, also acknowledge the reduced internal validity of these studies and their susceptibility to bias, emphasizing that results should be interpreted cautiously.
Remove the first four and last two paragraphs, which discuss financial, psychological, psychosocial, and palliative care aspects, as these are outside the scope of this systematic review and not supported by referenced data in the main tables.
Clarify what this review adds to previous meta-analyses and systematic reviews, including:
A more comprehensive overview of available evidence (despite its heterogeneity).
Differences in inclusion criteria, outcome definitions, or target populations.
More critical discussion of the quality and limitations of the available evidence.
Avoid using the discussion to support or advocate for one intervention, and instead focus on objective synthesis and identification of research gaps.
References Section
Please check all DOI links carefully. Many appear to be broken or improperly formatted. Use CrossRef or similar tools to verify and update each DOI to ensure proper access.
Figures and Tables
Figure 1 (Flow Diagram)
Include the box for studies excluded at the full-text stage, specifying the number and main reasons for exclusion (e.g., no intervention, wrong population, lack of outcome data).
Table 2A and 2B (Risk of Bias Assessment)
Replace current format with a color-coded risk of bias summary, using a scale similar to the Cochrane Risk of Bias Tool, with the following scheme:
Low risk (green)
Some concerns (yellow)
High risk (red)
This visual approach improves readability and interpretability for the reader.
Author Response
Comment 1: [Correct: “Fifty-seven studies with 655patients were included…”]
Response 1: [Thank you for your insightful feedback. Please find the addressed comments on page 2, lines 44.]
Comment 2: [Original: “and lack of standardized measurement methos drawing a firm conclusion is difficult.”]
Response 2: [Thank you for your insightful feedback. Please find the addressed comments on page 2, lines 59.]
Comment 3: [Remove the first sentence due to redundancy:
“Frontotemporal dementias are neurodegenerative illnesses that begin with symptoms of frontal and/or temporal lobe dysfunction.]
Response 3: [Thank you for your insightful feedback. The sentence has been removed.]
Comment 4: [Please clarify the outcome measures by naming the specific scores used in the included studies. For example, if available, indicate whether studies used standardized tools such as the Western Aphasia Battery or the Boston Naming Test. If these are unavailable, at least specify the type of data extracted, e.g., language performance metrics, qualitative descriptions, or pre/post therapy changes in communication.]
Response 4: [Thank you for your insightful feedback. Please find the addressed comments on pages 5-6, lines 190-199.]
Comment 5: [Correct final sentence:
“Ultimately, 57 studies involving 655patients met the inclusion criteria and were included in the review]
Response 5: [Thank you for your insightful feedback. Please find the addressed comments on pages 7, lines 236-237.]
Comment 6: [Please expand on how the term “experimental study” is defined by the authors. For example: Did it include any study involving an intervention with pre/post measures regardless of controls?. Were case series with intervention considered experimental, even if lacking a comparator group?. Clarify whether pilot trials, feasibility studies, or single-case experimental designs were classified under this term.]
Response 5: [Thank you for your insightful feedback. Please find the addressed comments on pages 8, lines 246-248.]
Comment 6: [It is crucial to clearly explain why this systematic review includes single-case designs, case reports, and case series, instead of limiting inclusion to controlled trials such as the 2025 meta-analysis by Lomi, which did not find evidence supporting the efficacy of speech and language therapy.
You should justify this methodological choice by addressing:
The limited number of RCTs in the field of frontotemporal dementia and speech therapy.
The exploratory or hypothesis-generating value of these lower-evidence studies.
The individualized nature of interventions in neurodegenerative diseases, which may lend themselves to single-case approaches.
However, also acknowledge the reduced internal validity of these studies and their susceptibility to bias, emphasizing that results should be interpreted cautiously.
Remove the first four and last two paragraphs, which discuss financial, psychological, psychosocial, and palliative care aspects, as these are outside the scope of this systematic review and not supported by referenced data in the main tables.
Clarify what this review adds to previous meta-analyses and systematic reviews, including:
A more comprehensive overview of available evidence (despite its heterogeneity).
Differences in inclusion criteria, outcome definitions, or target populations.
More critical discussion of the quality and limitations of the available evidence.
Avoid using the discussion to support or advocate for one intervention, and instead focus on objective synthesis and identification of research gaps.]
Response 6: [Thank you for your insightful feedback. Please find the addressed comments on pages 40, lines 660–671. We have discussed the inclusion of single-case studies and their limitations. The financial, psychological, psychosocial, and palliative care aspects were removed, and we now emphasize the strengths of our review and what it adds to previous meta-analyses and systematic reviews. Finally, no intervention was favored over another; the findings were presented according to the retrieved literature.]
Comment 7: [Please check all DOI links carefully. Many appear to be broken or improperly formatted. Use CrossRef or similar tools to verify and update each DOI to ensure proper access.]
Response 7: [ Thank you for your insightful feedback. All references have been checked, and the DOIs are correct.]
Comment 8: [Figure 1 (Flow Diagram)
Include the box for studies excluded at the full-text stage, specifying the number and main reasons for exclusion (e.g., no intervention, wrong population, lack of outcome data).
Table 2A and 2B (Risk of Bias Assessment)
Replace current format with a color-coded risk of bias summary, using a scale similar to the Cochrane Risk of Bias Tool, with the following scheme:
Low risk (green)
Some concerns (yellow)
High risk (red)
This visual approach improves readability and interpretability for the reader.]
Response 8: [Thank you for your insightful feedback. The exclusion reasons in Figure 1 have been provided as this is what we found in the literature and according to our search strategy. Regarding the risk of bias assessment tables, these are presented according to PRISMA guidelines and the well-known guidelines for risk of bias assessment.]
Reviewer 3 Report
Comments and Suggestions for Authors
Dear Author,
I appreciate te opportunity to read your paper !!
This is an interesting and well-designed work. I recommend the following
+I think it is appropiate to develop causes involved in PPA, as well as to include nutritional supplements and dietary considerations among current treatments….. mindfulness…
+ I don´t understand why Zdrugs are among the pharmacological treatments?? (Only 1 reference).
Author Response
Comment 1: [+I think it is appropiate to develop causes involved in PPA, as well as to include nutritional supplements and dietary considerations among current treatments….. mindfulness…]
Response 1: [Thank you for your insightful feedback. The causes and pathophysiology have already been discussed on page 4, lines 112–129, and pages 36-37, lines 547–586. Regarding the dietary considerations, there are no such considerations in the literature, as we have tried our best to address all relevant factors.]
Comment 2: [+ I don´t understand why Zdrugs are among the pharmacological treatments?? (Only 1 reference).]
Response 2: [Thank you for your insightful feedback. The drug discussed is based on the current literature according to the search strategy and eligibility criteria. We have not encountered any other differences addressing the usage of other drugs.]
Round 2
Reviewer 1 Report
Comments and Suggestions for Authors
Dear Authors,
I appreciate the improvements made to the article. Please re-check abbreviations and any mistakes or typos.
I understand your rationale for using a comprehensive review including all types of studies concerning various therapeutic strategies in PPA. As you have mentioned in chapter “Strengths and limitations” (the proper title should be: “Study strengths and limitations”) it confers reduced internal validity and bias susceptibility.
After your improvements, conclusions are consistent with the aims of the study.
Yours faithfully,
Comments on the Quality of English LanguageThe English could be improved, especially in the sentences formations.
Author Response
Comment 1: [I appreciate the improvements made to the article. Please re-check abbreviations and any mistakes or typos.]
Response 1: [Thank you for your insightful feedback. We assure you that all abbreviations and typographical errors have been addressed. Additionally, the manuscript has been thoroughly revised by three senior authors.]
Comment 2: [I understand your rationale for using a comprehensive review including all types of studies concerning various therapeutic strategies in PPA. As you have mentioned in chapter “Strengths and limitations” (the proper title should be: “Study strengths and limitations”) it confers reduced internal validity and bias susceptibility.]
Response 2: [Thank you for your insightful feedback. We have addressed the comment on page 40, line 658.]
Reviewer 2 Report
Comments and Suggestions for Authors
After the amedments now the paper is ready to be published, although references should be unified:
- year in brackets (refs 15 and 16, lines 769 and 772
- number of authors. Most of the references after three authors add et al, except:
Ref 9 (4)
Ref 15 (19)
Ref 16 (4)
Ref 23 (4)
Ref 26 (4)
Ref 44 (4)
Ref 48 (4)
Ref 52 (4)
Ref 71 (4)
Ref 74 (4)
Ref 85 (4)
Ref 99 (4)
Author Response
Comment 1: [
After the amedments now the paper is ready to be published, although references should be unified:
- year in brackets (refs 15 and 16, lines 769 and 772
- number of authors. Most of the references after three authors add et al, except:
Ref 9 (4)
Ref 15 (19)
Ref 16 (4)
Ref 23 (4)
Ref 26 (4)
Ref 44 (4)
Ref 48 (4)
Ref 52 (4)
Ref 71 (4)
Ref 74 (4)
Ref 85 (4)
Ref 99 (4)]
Response 2: [Thank you for your insightful feedback. We have addressed this issue and thoroughly revised the references. The references were automatically generated using citation software to ensure consistency throughout the manuscript.]